# Targeting the Tumor Microenvironment in EGFR-Mutant Lung Cancer: Opportunities and Challenges

**DOI:** 10.3390/biomedicines13020470

**Published:** 2025-02-14

**Authors:** Jeong Uk Lim, Junyang Jung, Yeon Wook Kim, Chi Young Kim, Sang Hoon Lee, Dong Won Park, Sue In Choi, Wonjun Ji, Chang Dong Yeo, Seung Hyeun Lee

**Affiliations:** 1Division of Pulmonary and Critical Care Medicine, Department of Internal Medicine, Yeouido St. Mary’s Hospital, College of Medicine, The Catholic University of Korea, Seoul 06591, Republic of Korea; 2Department of Anatomy and Neurobiology, College of Medicine, Kyung Hee University, Seoul 02447, Republic of Korea; 3Division of Pulmonary and Critical Care Medicine, Department of Internal Medicine, Seoul National University Bundang Hospital, Seongnam 13620, Republic of Korea; 4Division of Pulmonology, Department of Internal Medicine, Yonsei University College of Medicine, Seoul 03722, Republic of Korea; 5Division of Pulmonary and Critical Care Medicine, Department of Internal Medicine, Institute of Chest Diseases, Severance Hospital, Yonsei University College of Medicine, Seoul 03722, Republic of Korea; 6Division of Pulmonary Medicine and Allergy, Department of Internal Medicine, Hanyang University College of Medicine, Seoul 04763, Republic of Korea; dongwonpark@hanyang.ac.kr; 7Division of Pulmonary, Allergy and Critical Care Medicine, Department of Internal Medicine, Korea University College of Medicine, Seoul 02841, Republic of Korea; 8Division of Pulmonology and Critical Care Medicine, Department of Internal Medicine, Asan Medical Center, University of Ulsan College of Medicine, Seoul 44610, Republic of Korea; 9Division of Pulmonary, Critical Care and Sleep Medicine, Department of Internal Medicine, Eunpyeong St. Mary’s Hospital, College of Medicine, The Catholic University of Korea, Seoul 03083, Republic of Korea; 10Division of Pulmonary, Allergy, and Critical Care Medicine, Department of Internal Medicine, College of Medicine, Kyung Hee University, Seoul 02447, Republic of Korea; 11Department of Precision Medicine, Graduate School, Kyung Hee University, Seoul 02447, Republic of Korea

**Keywords:** epidermal growth factor receptor mutation, lung cancer, resistance, tumor microenvironment, biomarker

## Abstract

Tyrosine kinase inhibitors (TKIs) have transformed the treatment of epidermal growth factor receptor (EGFR)-mutant non-small cell lung cancer. However, treatment resistance remains a major challenge in clinical practice. The tumor microenvironment (TME) is a complex system composed of tumor cells, immune and non-immune cells, and non-cellular components. Evidence indicates that dynamic changes in TME during TKI treatment are associated with the development of resistance. Research has focused on identifying how each component of the TME interacts with tumors and TKIs to understand therapeutic targets that could address TKI resistance. In this review, we describe how TME components, such as immune cells, fibroblasts, blood vessels, immune checkpoint proteins, and cytokines, interact with EGFR-mutant tumors and how they can promote resistance to TKIs. Furthermore, we discuss potential strategies targeting TME as a novel therapeutic approach.

## 1. Introduction

Lung cancer is the most fatal form of cancer worldwide, with 2.2 million new diagnoses and 1.79 million deaths annually. It is primarily categorized into two histological types: non-small cell lung cancer (NSCLC) and small cell lung cancer (SCLC), with NSCLC accounting for 85% of the cases [1]. NSCLC, comprising adenocarcinoma, squamous cell carcinoma, and large cell carcinoma, presents diverse molecular profiles and treatment responses [2]. Recent advancements in targeted therapies have substantially improved the outcomes of patients with advanced NSCLC harboring driver genetic alterations. Epidermal growth factor receptor (EGFR) is the most common targetable driver gene, with mutation incidence of up to 60% in Asian NSCLC populations, which is significantly higher than that in Western populations (10–15%) [3,4]. Targeted therapies have significantly advanced the treatment of NSCLC by selectively inhibiting oncogenic driver mutations, leading to improved patient outcomes (Table 1) [5]. Various clinical trials and real-world studies have demonstrated the survival benefits of EGFR tyrosine kinase inhibitors (TKIs) in advanced disease. Recently, a newer generation EGFR-TKI, osimertinib, has been approved as an adjuvant treatment for early-stage EGFR-mutant lung cancer after curative resection [6]. Furthermore, novel combinational approaches, such as combining EGFR-TKIs with chemotherapy or an EGFR mesenchymal-epithelial transition factor (MET) bispecific antibody, have recently shown better clinical efficacy than EGFR-TKI monotherapy and emerged as new frontline treatment options [7,8]. However, resistance to TKIs eventually develops via conformational changes in target proteins or activation of bypass signaling pathways [9,10].

The tumor microenvironment (TME) plays a critical role in carcinogenesis and treatment of lung cancer, influencing various aspects of tumor development, progression, and therapeutic outcomes. In lung cancer carcinogenesis, the TME contributes significantly to chronic inflammation, inducing tissue injury and repair cycles [11]. This inflammatory microenvironment promotes bronchioalveolar stem cell growth and activates key signaling pathways such as nuclear factor-kappa B and signal transducer and activator of transcription 3 (STAT3), which are essential for lung cancer development [11,12]. The TME also facilitates immune escape, tumor angiogenesis, and epithelial-mesenchymal transition (EMT), which contribute to cancer progression and metastasis [12,13]. In addition, hypoxia and nutrient deprivation in the TME promote cancer progression by activating angiogenesis, inducing resistance to apoptosis, and selecting resistant clones [14]. Complex interactions between tumors and various cell types, such as macrophages, neutrophils, and cancer-associated fibroblasts (CAFs), significantly affect the clinical outcomes of cancer treatment, particularly immunotherapy [14,15,16,17]. Although the clinical implications of TME in oncogene-addicted lung cancers have not been fully understood, recent evidence demonstrates that those tumors actively interact with the surrounding TME. Further, the TME is influenced by TKI treatment, which may be associated with the development of drug resistance [18]. In this review, we describe the roles of various TME components, such as immune cells, fibroblasts, blood vessels, immune checkpoint proteins, and cytokines, and their involvement in promoting resistance to TKIs in EGFR-mutant lung cancer. We also discuss the potential strategies to overcome TME-associated resistance and the rationale for novel treatments targeting TME or using immunomodulation.

**Table 1 biomedicines-13-00470-t001:** Current FDA-approved targeted therapy in advanced NSCLC.

Tyrosine Kinase Inhibitor	Therapeutic Target	Reference
Gefitinib	EGFR (Exon 19 deletions, L858R mutations)	[19]
Erlotinib	EGFR (Exon 19 deletions, L858R mutations)	[20]
Afatinib	EGFR (Exon 19 deletions, L858R mutations, uncommon mutations like G719X, L861Q, S768I)	[21]
Osimertinib	EGFR (Exon 19 deletions, L858R mutations, T790M)	[22]
Lazertinib	EGFR (Exon 19 deletions, L858R mutations, T790M)	[23]
Dacomitinib	EGFR	[24]
Alectinib	ALK	[25]
Ceritinib	ALK, ROS1	[26]
Brigatinib	ALK	[27]
Lorlatinib	ALK, ROS1	[28]
Crizotinib	ALK, ROS1, MET	[29]
Capmatinib	MET exon 14 skipping mutations	[30]
Tepotinib	MET exon 14 skipping mutations	[31]
Selpercatinib	RET	[32]
Pralsetinib	RET	[33]
Amivantamab	EGFR, MET	[34]
Mobocertinib	EGFR exon 20 insertion mutations	[35]
Dabrafenib + Trametinib	BRAF V600E	[36]
Encorafenib + Binimetinib	BRAF V600E	[37]

Abbreviations: EGFR, epidermal growth factor receptor; ALK, anaplastic lymphoma kinase; ROS1, c-ros oncogene 1; MET, mesenchymal-epithelial transition factor; RET, rearranged during transfection; BRAF, v-Raf murine sarcoma viral oncogene homolog B; V600E, valine-to-glutamate substitution at codon 600.

## 2. Mechanism of Resistance to EGFR-TKIs: An Overview

The emergence of resistance to EGFR TKIs poses a significant challenge in the treatment of EGFR-mutant NSCLC. EGFR mutations play a critical role in the pathogenesis of NSCLC by driving aberrant activation of tyrosine kinase signaling pathways. These mutations lead to the constitutive activation of the receptor’s intrinsic tyrosine kinase domain, independent of ligand binding [38]. The sustained activation leads to the activation of the MAPK, AKT, STAT3, and other downstream oncogenic signaling pathways, promoting processes such as uncontrolled cell proliferation, inhibition of apoptosis, angiogenesis, and metastasis [39]. EGFR-TKI resistance can be categorized into two types, i.e., primary and secondary (acquired) resistance. Primary resistance occurs in patients who do not respond to EGFR-TKIs during frontline treatment or experience early relapse within 6 months. This can be further divided into intrinsic and late primary resistance [40]. The mechanisms underlying primary resistance are not fully understood but may involve pre-existing genomic alterations, including de novo T790M and TP53 mutations, and MET amplification [40].

Secondary or acquired resistance typically develops after an initial response to EGFR-TKIs. This type of resistance develops via either on-target or off-target mechanisms. On-target resistance mechanisms include alterations in the target enzyme, typically the TKI-binding tyrosine kinase, which results in a reduction in the binding affinity of TKIs, making them less capable of inhibiting kinase activity [41]. The most frequent form of on-target resistance is the acquisition of secondary mutations in the gene encoding the target kinase, e.g., the T790M or C797X mutations [42]. For example, T790M mutations alter the configuration of the ATP-binding site of the tyrosine kinase domain, reducing the binding affinity of TKIs and accounting for approximately 50% of the acquired resistance after first- and second-generation EGFR-TKI treatment [43].

Off-target resistance involves changes in cellular pathways or molecules other than the primary target of TKIs, accounting for 10–15% and 30% resistance to early- and third-generation EGFR-TKIs, respectively [44,45,46,47]. These mechanisms include activation of alternative pathways (such as c-MET and AXL), aberrations in downstream pathways (such as phosphatase and tensin homolog loss), and TKI-mediated apoptosis impairment (such as Bcl-2–like 11 (BIM) deletion or polymorphism) [48]. In addition, histological transformations to squamous cell or small cell lung cancer and EMT are also rare but clinically significant resistance mechanisms.

## 3. Current Treatment Strategy for Overcoming EGFR TKI Resistance with Targetable Co-Mutations

The current approach to overcoming EGFR TKI resistance emphasizes managing actionable co-mutations. Extensive clinical efforts aim to identify these alterations at the time of resistance. However, for patients without detectable genetic changes that allow for targeted intervention, treatment options remain limited. Some key genetic alterations that serve as targets for overcoming EGFR TKI resistance will be discussed here.
**C797X mutation**

Upon progression on osimertinib, approximately 15% of tumors develop on-target mutations [49], with EGFR C797X in exon 20 being the most prevalent. This mutation hampers the covalent binding of osimertinib to the EGFR kinase domain. Other notable acquired mutations include L718Q/V, G719A, and G724S in exon 18 [46]. Fourth-generation EGFR TKIs like BLU-94573 and BBT-17674 have been developed, showing promising initial data. However, further investment in BLU-945 for EGFR-mutant NSCLC has been discontinued, and new treatments are awaited. Additionally, preclinical studies suggest that cancer cells with the acquired C797S mutation after osimertinib therapy remain sensitive to 1G or 2G EGFR TKIs [50]. A multicenter, open-label, phase 1/2 trial (NCT05394831) is investigating JIN-A02, a fourth-generation EGFR-TKI. The results may provide insights into its potential as a treatment option for advanced NSCLC patients with C797S and/or T790M mutations [51]. Another fourth-generation TKI, BDTX-1535, showed promising outcomes in patients with refractory or relapsed EGFR-mutant NSCLC in a phase 2 trial (NCT05256290) [52].
***MET alteration***

In NSCLC, MET-dependent resistance emerges as a significant obstacle, often activated by the formation of homodimers or through trans-activation by other tyrosine kinase receptors. MET amplification contributes to resistance in approximately 50–60% of cases treated with first- and second-generation EGFR TKIs [53,54], and in 15–19% of cases involving third-generation EGFR TKIs [55]. Overcoming this resistance requires the concurrent targeting of both EGFR and MET receptors, highlighting the potential utility of anti-MET agents in combination with EGFR TKIs to achieve a more effective antitumor response [56].

To overcome EGFR TKI resistance combined with MET amplification, a combinatorial approach of EGFR TKI and crizotinib has been attempted [57,58]. Given the limited antitumor efficacy of MET TKIs as monotherapy (ORR: 8.3%) in addressing acquired MET amplification, the combination of a MET TKI with an EGFR TKI has emerged as the most effective strategy to date [59]. This dual inhibition strategy, combined with osimertininb, is being explored in trials such as SAVANNAH (savolitinib) [60], INSIGHT2 (tepotinib) [61], and ORCHARD (savolitinib) [62], which have reported ORRs up to 50% and a median PFS of 5.0 months. Although this approach might improve outcomes compared to standard platinum-pemetrexed chemotherapy [63], its efficacy needs confirmation through ongoing trials like GEOMETRY-E (NCT04816214) and SAFFRON (NCT05261399). For osimertinib-relapsed, chemotherapy-naïve EGFR-mutant NSCLC patients, the combination of amivantamab and lazertinib has shown promising results, especially in tumors with MET overexpression by immunohistochemistry [64]. A recent update from the INSIGHT-2 study (NCT03940703), an open-label, phase 2 trial, reported that the combination therapy of tepotinib 500 mg and osimertinib 80 mg daily achieved a significant ORR of 50.0% (95% CI 39.7–60.3) in patients with MET amplifications who had progressed following initial osimertinib treatment [65].


**
*HER2 and HER3 alterations*
**


Human epidermal growth factor receptor 2 (HER2) is a tyrosine kinase receptor encoded by the ERBB2 gene [66]. HER2 amplification is observed in 5% of patients who develop resistance to second-line osimertinib and in 2% of cases using first-line osimertinib [67,68]. Distinct from HER2 amplification, HER2 mutations are considered to be more relevant to lung carcinogenesis, and are detected in approximately 2–4% of patients with NSCLC patients [69,70]. Furthermore, both HER2 amplification and mutation are usually mutually exclusive with other targetable mutations [71,72].

One approach to overcoming HER2 amplification implicated in osimertinib resistance in EGFR-mutant NSCLC involves the combination of trastuzumab-emtansine and osimertinib. In a multicenter, single-arm, phase 1–2 study (NCT03784599), patients treated with this regimen showed a limited ORR of 4% and a median progression-free survival (PFS) of 2.8 months, indicating the need for alternative strategies [73].

HER3 is a member of the EGFR family, and heterodimerizes with other HER proteins. It is involved in cancer cell proliferation by downstream signaling through the PI3K/protein kinase B (AKT) pathway [74]. HER3, when combined with receptors such as MET, HER2, and EGFR, becomes a potent signaling entity, producing strong growth signals that can enhance resistance to targeted therapeutic interventions [75].

An antibody-drug conjugate (ADC) is a class of drug that consists of a monoclonal antibody linked to a cytotoxic payload via a stable chemical linker, enabling targeted delivery of the drug to cancer cells [76]. Initial studies indicate significant potential for ADCs in treating osimertinib-resistant EGFR-mutant NSCLC. This includes ADCs targeting HER3, such as patritumab deruxtecan and BL-B01D1 (a bispecific ADC targeting EGFR and HER3), ADCs targeting TROP2 like datopotamab deruxtecan, and ADCs targeting cMET such as telisotuzumab vedotin [77,78,79,80]. The HERTHENA-Lung02 study (NCT05338970), which included about 560 patients with EGFR mutations who had progressed during EGFR TKI therapy, indicated that patritumab deruxtecan might be effective in treating the EGFR TKI-resistant subgroup [81]. In contrast to the other three agents, telisotuzumab vedotin requires high MET expression for efficacy, defined as a c-Met expression level of 3+ in at least 25% of tumor cells [80].


**Other targetable mutations (RET, BRAF, PIK3CA)**


RET fusions are also reported as acquired resistance mechanism after EGFR TKI treatment [82,83]. When compared to primary RET fusions, the proportion of CCDC6-RET in patients with acquired resistance to EGFR TKIs was higher. Additionally, RET fusions were more frequently linked to acquired resistance to third-generation EGFR-TKIs compared to earlier generations [84]. Fourteen patients who showed acquired RET fusions after osimertinib treatment underwent osimertinib and selpercatinib, showing modest response rate, disease control rate, and median treatment duration, were recorded at 50% (95% CI: 25–75%, n = 12), 83% (95% CI: 55–95%), and 7.9 months, respectively [85]. For BRAF V600E-mediated osimertinib resistance, combinations such as dabrafenib and trametinib with osimertinib, as well as vemurafenib with osimertinib, have been reported in case studies involving patients resistant to osimertinib [86,87].

Recent studies indicate that the activation of the PI3K/AKT/mTOR signaling pathway contributes to the aggressive nature of lung cancer [88]. Currently, there are no established treatments that effectively target both the EGFR and PI3K/AKT/mTOR pathways simultaneously. Combinatorial approaches such as PI3K inhibitors with EGFR TKIs and PI3K/mTOR inhibitors with EGFR TKIs in overcoming EGFR TKI resistance mediated by the PI3K/AKT/mTOR pathway are to be investigated [89].

## 4. Role of the TME in EGFR-TKI Resistance

The TME is a dynamic and intricate system consisting of tumor cells, adjacent immune and non-immune cells, and non-cellular components [90]. According to the traditional ‘seed and soil’ theory, tumor cells require a suitable surrounding environment. To escape immune surveillance and survive, tumor cells utilize immune cells and non-cellular components to create an immunosuppressive environment [91]. In addition, the TME is further remodeled during TKI treatment through various components such as tumor-infiltrating immune cells, immune-modulating molecules, cytokines, and chemokines, which are collectively associated with the development of drug resistance [92]. Owing to the critical role of the TME in determining immunotherapy efficacy, it is necessary to understand the important modifications within the TME after TKI treatment, and how these changes contribute to TKI resistance [92] (Table 2, Figure 1). Furthermore, modifying components of the TME can help restore drug sensitivity, delay resistance, and enhance treatment efficacy [93]. The TME, composed of immune cells, stromal cells, and extracellular matrix (ECM) components, plays a critical role in tumor progression and therapeutic response. For instance, targeting immune-suppressive factors such as tumor-associated macrophages (TAMs) or regulatory T cells can improve the effectiveness of targeted therapies and even synergize with immunotherapy [94]. Additionally, tumor modulation strategies allow for the co-targeting of alternative pathways that contribute to resistance against EGFR inhibitors.

**Table 2 biomedicines-13-00470-t002:** Explanation of each component of the TME in EGFR-mutant NSCLC.

Components	EGFR-Mutant State	Potential Role as a Biomarker and Strategic Approaches	References
CD8+ lymphocytes	CD8+ T-cell infiltration increases in EGFR-mutant NSCLC responsive to TKI but decreases in resistant tumors.	Lower levels of CD8+T-cell activity are linked to resistance in EGFR-mutant NSCLC.	[95,96,97,98,99,100,101]
CD4+ lymphocytes	Regulatory T (FOXP3+) cells (Tregs) are increased in EGFR-mutant NSCLC, contributing to an immunosuppressive microenvironment.	High Treg (FOXP3+) counts in EGFR TKI–resistant tumors are linked to immune suppression and resistance.	[98,102,103,104,105,106,107]
PD-(L)1	PD-L1 expression often increases in EGFR TKI–resistant NSCLC, contributing to immune escape.	Elevated PD-L1 levels after EGFR TKI resistance can indicate worse prognosis. Targeting PD-L1 or related pathways (STAT3, ERK1/2) may enhance response in EGFR TKI–resistant NSCLC.	[98,108,109,110,111,112,113,114,115,116,117,118]
TAM	In EGFR-mutant NSCLC, M2-like macrophages support tumor progression, increasing in TKI-resistant tumors.	Increased M2 macrophages are linked to EGFR-TKI resistance. Macrophage polarization (M2 to M1) may predict improved responses in EGFR-TKI-resistant NSCLC. Combining STING agonists or liposome therapies with EGFR TKIs reprograms TAMs, enhancing tumor regression.	[100,103,119,120,121,122,123]
Cytokines	Increased levels of cytokines such as IL-6, IL-8, and TGF-β promote EGFR TKI resistance and tumor progression.	Elevated cytokines (IL-22, IL-6) in plasma and tumor tissue are associated with resistance to EGFR-TKI therapy. Targeting cytokines (IL-6, IL-8, TGF-β) combined with EGFR-TKIs may help overcome resistance in NSCLC.	[92,93,124,125,126,127,128,129,130,131]
Exosomes	Exosomes from EGFR-mutant NSCLC carry tumor-related RNA, contributing to EGFR-TKI resistance through signaling pathways.	Exosomal miRNA and EGFR expression serve as potential biomarkers for predicting resistance. Modulating exosomal signaling pathways such as PI3K/AKT and ERK1/2 may overcome TKI resistance.	[103,132,133,134,135,136,137,138,139,140,141,142,143,144,145,146,147]
CAF	CAFs drive resistance in EGFR-mutant NSCLC by promoting EMT and secreting resistance-inducing factors such as HGF, IL-6, and kynurenine, activating pro-survival pathways in cancer cells.	CAF markers such as α-SMA, HGF, and podoplanin in tumor tissues could predict EGFR-TKI resistance. Targeting CAF-derived factors such as HGF or using antifibrotic agents may counteract TKI resistance.	[148,149,150,151,152,153,154,155,156,157,158,159,160,161,162,163]
Vasculature (VEGF)	VEGF upregulation in EGFR-mutant NSCLC cells contributes to TKI resistance, promoting angiogenesis and tumor progression.	High VEGF/VEGFR-2 expression correlates with poor outcomes in EGFR-mutant NSCLC. Combining VEGF inhibitors such as bevacizumab or ramucirumab with EGFR-TKIs significantly improves PFS. Anlotinib has shown improved survival outcomes in TKI-resistant patients when combined with immune checkpoint inhibitors.	[164,165,166,167,168]

Abbreviations: EGFR: epidermal growth factor receptor; NSCLC: non-small cell lung cancer; TKI: tyrosine kinase inhibitor; TME: tumor microenvironment; CD8+: cluster of differentiation 8; CD4+: cluster of differentiation 4; PD-1: programmed death-1; PD-L1: programmed death-ligand 1; TAM: tumor-associated macrophage; M1: classically activated macrophage (anti-tumor); M2: alternatively activated macrophage (pro-tumor); STING: stimulator of interferon genes; IL-6: interleukin 6; IL-8: interleukin 8; TGF-β: transforming growth factor beta; CAF: cancer-associated fibroblast; HGF: hepatocyte growth factor; α-SMA: alpha smooth muscle actin; EMT: epithelial-to-mesenchymal transition; VEGF: vascular endothelial growth factor; VEGFR: vascular endothelial growth factor receptor; FGFR: fibroblast growth factor receptor; PDGFR: platelet-derived growth factor receptor; PFS: progression-free survival; EOMES: eomesodermin.

### 4.1. Cellular Components

#### 4.1.1. Cluster of Differentiation 8 (CD8+) Lymphocytes

CD8+ T-cells exert direct cytotoxic effects on cancer cells by secreting granzymes and perforins. They are activated by tumor-resident antigen-presenting cells or MHC class I molecules on tumor cells, leading to tumor cell destruction and neoantigen release, which can initiate secondary immune responses [95,96].

Fang et al. reported that TKI treatments significantly enhanced immune cell infiltration and cytotoxic activity in samples responsive to TKI, while no such improvements were observed in samples resistant to the therapy [97]. Another study using paired baseline and re-biopsy samples after progression from EGFR-TKI therapy showed that CD8+ and FoxP3+ tumor-infiltrating lymphocyte densities decreased following EGFR-TKI treatment [98]. Another study reported a significant decrease in the number of CD8+ T-cells in tumors after the development of resistance to EGFR-TKIs [99]. In addition, single-cell RNA sequencing revealed that increased infiltration of CD8+ T-cells correlated with a positive response to EGFR-TKI treatment, whereas decreased infiltration correlated with treatment resistance [100].

Although an overall reduction in CD8+ T-cell infiltration has been linked to EGFR-TKI resistance, a specific subset of CD8+ T-cells may be correlated with this resistance. Notably, an increase in eomesodermin (EOMES)-positive CD8+ T-cells, a marker associated with the thymic precursors of self-specific memory-phenotype CD8+ T-cells and immune homeostasis, has been linked to EGFR-TKI resistance [169]. Further analysis confirmed elevated EOMES+ CD8+ T-cell counts in both the tissue and peripheral blood from patients exhibiting resistance to TKIs [169].

#### 4.1.2. CD4+ Lymphocytes

CD4+ T-cells play a central role in coordinating adaptive immune responses by facilitating the recruitment and activation of dendritic cells, crucial for CD8+ T-cell priming [102]. Five main subsets of CD4+ T helper cells have been identified: Th1, Th2, Th17, regulatory T-cells (Treg), and follicular helper T-cells [103,104].

Among various subsets of CD4+ cells, the role of Tregs in EGFR-TKI resistance is relatively well-known. Tregs, generally characterized by the expression of the transcription factor FoxP3, play an important role in maintaining immune tolerance and modulating the immune response [105]. The TME of EGFR-mutant tumors is more likely to have an increased count of Treg cells than that of the wild type [106]. An in vitro study using the EGFR-mutant lung cancer cell lines HCC827 and H4006 co-cultured with activated peripheral blood mononuclear cells and treated with EGFR-TKIs showed notable changes in immune cell dynamics. Treg proportions increased significantly after co-culture and remained unchanged during EGFR-TKI treatment, suggesting their potential role in TKI resistance [170]. In a study using paired tumor samples, the density of Tregs shifted from negative or low at baseline to high after the development of EGFR-TKI resistance in programmed death-ligand 1 (PD-L1)-negative or -positive tumors, indicating an immunosuppressive role of Tregs in oncogene-driven tumors [98].

#### 4.1.3. Tumor-Associated Macrophages (TAMs)

Macrophages are the most abundant innate immune cells in the TME and are commonly referred to as TAMs. The phenotypes and functions of TAMs are heterogeneous and can change dynamically within the surrounding microenvironment. Unpolarized M0 macrophages can be polarized into two functional states: pro-inflammatory M1 macrophages (classically activated) and pro-tumorigenic M2 macrophages (alternatively activated) [171]. These subtypes exhibit high plasticity, allowing them to switch between different forms or alter their functions in the TME [171,172]. M1 macrophages typically exert anti-tumor functions, including direct cytotoxicity and antibody-dependent cell-mediated cytotoxicity to kill tumor cells [173]. In contrast, M2 macrophages promote tumor occurrence, metastasis, angiogenesis, and progression while inhibiting T-cell-mediated anti-tumor immune responses by secreting anti-inflammatory cytokines and immune-suppressive factors [173,174]. In addition, a retrospective study indicated that increased TAM infiltration into tumors was associated with worse clinical outcomes in EGFR-mutant patients treated with EGFR-TKIs [175],

A preclinical study using functional mouse models demonstrated that TAM influx into the lungs precedes the development of a progenitor-like cell state in EGFR-mutant lung alveolar type II epithelial cells, suggesting a potential link between macrophage activity and tumorigenesis in EGFR-mutant cells [120]. A clinical study, using single-cell RNA sequencing of biopsy specimens from EGFR-mutant NSCLC, showed that macrophage infiltration decreased with EGFR-TKI treatment, to which the tumor was sensitive and increased in resistant cases. Upon disease progression, the infiltration of IDO1+ macrophages, proliferative regulatory T-cells, and dysfunctional T-cells, which are scarcely observed in residual diseases, was noted. In contrast, the absence of a TKI response was marked by an increase in immunosuppressive M2-like macrophages [100]. This was further supported by a study by Han et al., in which tissues resistant to osimertinib exhibited an increase in M2 macrophages, indicating enhanced macrophage polarization [103]. In this study, M2 polarization was mediated by tumor-derived exosomes, highlighting the critical role played by exosomes in the development of TKI resistance.

#### 4.1.4. CAFs

CAFs are the primary producers of collagen and directly interact with various stromal cells, including endothelial and inflammatory cells. CAFs have been attributed to several cell types, including locally infiltrating fibroblasts, endothelial cells, pericytes, vascular adventitial fibroblasts, and cancer cells that undergo fibroblastic changes during EMT [148]. Multiple markers, such as α-smooth muscle actin (α-SMA), tenascin-C, platelet-derived growth factor receptor (PDGFR)-A/B, fibroblast activation protein-α, CD90, and podoplanin, are used to identify CAFs. Although α-SMA-positive fibroblasts are commonly known as myofibroblasts, all CAFs do not express α-SMA [148].

As essential stromal components of the TME [149], CAFs play a central role in the development of EGFR-TKI resistance. When EGFR TKI-sensitive NSCLC cells are co-cultured with CAFs, the cells become resistant to TKIs, suggesting that resistance may develop through direct contact [150]. CAFs significantly contribute to lung cancer resistance to EGFR-TKIs by inducing EMT through specific CAF-mediated signaling pathways [151]. CAFs are heterogeneous, with various subtypes distinguished by markers such as vimentin, α-SMA, podoplanin, and periostin [152,153].

In addition, CAFs produce humoral factors that contribute to EGFR-TKI resistance. Suzuki et al. identified CAF-derived hepatocyte growth factor (HGF) as an important factor in inducing EGFR-TKI resistance. A study using an EGFR-mutant cell line PC-9, treated with conditioned media from CAFs and their matched non-cancerous tissue-associated fibroblasts, suggested that EGFR-TKI resistance-promoting CAFs are marked by the elevated secretion levels of HGF. This was demonstrated by the effects observed with the use of an HGF-neutralizing antibody [154]. In a previous study involving mouse models and human samples, proteins produced by CAFs from tumor tissues, such as kynurenine, activated the downstream AKT and extracellular signal-regulated kinase (ERK) pathways, contributing to EGFR-TKI resistance [155]. In a study by Tan et al., a lung-on-a-chip model was employed to co-culture NSCLC cell lines with human fibroblasts and endothelial cells to create a simulated TME for examining EGFR-TKI resistance mechanisms. Human fetal lung fibroblasts transform into CAFs and IL-6 facilitates this transformation, inducing EMT in NSCLC cells, which contributes to their resistance to osimertinib [156].

Evidence suggested that certain CAF subtypes play a dominant role in resistance to EGFR-TKIs. In a study using a xenograft model, a CAF subpopulation enriched in IL-6 production was the predominant CAF type in drug-tolerant persister cells that emerged after repeated erlotinib treatment [157]. Additionally, podoplanin-positive CAFs have been implicated in EGFR TKI resistance in EGFR-mutant lung adenocarcinoma, as evidenced by in vitro studies showing that cancer cells co-cultured with these CAFs exhibit increased resistance [150]. Moreover, adenocarcinomas with solid-predominant histology, which tend to exhibit high primary resistance to EGFR-TKIs, frequently contain more podoplanin-positive CAFs than other histological variants [158,159]. In contrast, a study using patient-derived CAFs demonstrated that CAF heterogeneity contributes to overall TKI resistance. We found that MET and/or fibroblast growth factor receptor (FGFR) activation by CAFs could repeatedly rescue EGFR-mutant cancers [160].

### 4.2. Non-Cellular Components

#### 4.2.1. Cytokines

Cytokines are crucial TME components and activate or suppress immune responses against cancer cells. The cytokine levels in the TME are also affected by EGFR-TKI treatment [92]. Generally, immunosuppressive cytokines are upregulated, whereas immune-promoting factors are downregulated in EGFR-mutant tumors [92]. Evidence has consistently reported that elevated cytokine levels in the plasma and/or tumor samples are associated with TKI resistance.

In one study, IL-22 levels were higher in the EGFR TKI-resistant group than in the EGFR TKI-sensitive group. These levels were also associated with EGFR TKI efficacy [124]. Another study revealed that elevated CXCL10 levels during early EGFR-TKI treatment enhance oncogenic signaling and promote EGFR-TKI resistance through autocrine and paracrine pathways [125].

IL-6 is a known activator of the Janus kinase/STAT3 pathway and is involved in lung cancer metastasis [176]. In a study involving osimertinib-resistant cell lines and xenograft models, IL-6 levels were elevated. This study further demonstrated that IL-6 inhibition could reverse this resistance [126]. In a study using a genetically engineered mouse model of EGFR-mutant NSCLC, tumors resistant to EGFR-TKIs showed a notable increase in IL-6 secretion. The reduction in IL-6 was associated with an increased activity of infiltrating natural killer (NK) and T-cells and a decrease in Treg and Th17 cell populations [127].

IL-8 contributes to angiogenesis, cancer cell growth and survival, tumor cell motility, and leukocyte infiltration, which promote tumor progression [177]. In NSCLC, elevated IL-8 levels are associated with treatment resistance, with data from *The Cancer Genome Atlas* (*TCGA*) dataset showing a negative prognostic impact of high IL-8 levels in lung cancer, particularly in patients treated with chemotherapy and/or immunotherapy [128,178]. In EGFR-mutant lung cancer, IL-8 is upregulated in gefitinib-resistant cells, and high plasma IL-8 levels are correlated with shorter progression-free survival (PFS) in patients treated with EGFR-TKIs [179]. Moreover, IL-8 suppression enhanced gefitinib-induced cell death in gefitinib-resistant cells by inducing the loss of stem cell-like characteristics, suggesting a potential role of IL-8 in EGFR-TKI resistance, consistent with chemotherapy and immunotherapy findings [179]. In another study, blocking IL-8 signaling effectively reduced the mesenchymal features of TKI-resistant cells and markedly enhanced their susceptibility to erlotinib, verifying that IL-8 is a potential therapeutic target for overcoming EGFR-TKI resistance [129].

Transforming growth factor-beta (TGF-β) plays a complex and multifaceted role in cancer development and treatment [130]. TGF-β acts as a tumor suppressor in the early stages of cancer, inhibiting cell growth and promoting apoptosis [180]. However, as tumors progress, it promotes tumor growth, invasion, and metastasis [181]. This dual role of TGF-β complicates its relationship with EGFR-TKI treatment, as the cancer stage and specific tissue context can influence its effects. The TGF-β signaling pathway may contribute to TKI resistance by promoting tumor cell survival and metastasis in the later stages of cancer [181]. Additionally, both EGFR and TGF-β pathways are involved in regulating the TME and immune responses, which can impact treatment efficacy [180,182]. TGF-β2 was upregulated in osimertinib-resistant cells, leading to EMT and mothers against decapentaplegic homolog 2 pathway activation [131].

#### 4.2.2. Exosomes

Exosomes are extracellular vesicles, 30–150 nm in diameter, produced via the endoplasmic reticulum pathway. They function as carriers, transporting an array of substances such as proteins, nucleic acids, metabolites, and lipids originating from diverse cells [132], and play a pivotal role in cell-to-cell communication [133,134]. Lung cancer cells produce significant amounts of extracellular vesicles, predominantly as exosomes [135]. Normal human blood contains approximately 2000 trillion exosomes; however, this number is significantly increased in patients with cancer, with counts reaching approximately 4000 trillion [136]. These exosomes play crucial roles in various immune functions, including antigen presentation, immune activation and suppression, surveillance, and cell communication [137].

Especially, exosomal miRNAs may also play an important role in predicting treatment resistance and modulating the TME. The miRNA content in circulating exosomes closely resembles that in the originating cancer cells, suggesting their potential application in cancer diagnostics [140]. Furthermore, detecting exosomal miRNAs in body fluids underscores their promising role as disease biomarkers to predict treatment resistance [141]. miRNAs, abundantly present in exosomes, can modulate the stability or translation of target mRNAs, significantly altering the TME in lung cancer [142]. Liu et al. revealed that treatment with exosomes released by EGFR TKI-resistant lung cancer cells resulted in the acquisition of resistance in TKI-sensitive cells via phosphoinositide 3-kinase (PI3K)/AKT signaling pathway activation, suggesting a potential contribution of exosomes to the emergence of TKI resistance [143].

Notably, exosomes are critically involved in the polarization of macrophages into the M2 phenotype, which plays an immunosuppressive role in the TME [103]. Additionally, exosomes extracted from TAMs promote TKI resistance in lung cancer cells through AKT, ERK1/2, and STAT3 signaling pathway reactivation [144]. In a study evaluating the mechanism of action of exosome-derived miRNAs in osimertinib resistance, exosome-derived miR-184 and miR-3913-5p expression was significantly increased in the blood of patients with osimertinib resistance, suggesting that specific miRNA signatures are involved in the development of TKI resistance and can be used to predict resistance [145].

#### 4.2.3. VEGF and Vasculature

Angiogenesis is a key feature of tumors as it helps provide a continuous supply of nutrients. This process is controlled by several growth factors, with VEGF as a key mediator [164,165]. VEGF is often upregulated in response to hypoxic conditions within tumors [183,184], stimulating endothelial cell proliferation and new capillary formation to meet the metabolic demands of expanding cancer cells [185,186]. Interestingly, VEGF shares common downstream signaling with the EGFR pathway and activated EGFR signaling can upregulate VEGF expression [187]. Furthermore, a preclinical study showed that VEGF levels were significantly higher in EGFR-mutant cells than in wild-type cells [188].

In addition to its role in cancer cell proliferation, VEGF-driven angiogenesis contributes to an immunosuppressive TME [189,190]. Increased VEGF expression facilitates the recruitment of immune cells, such as TAMs and Tregs, which negatively regulate anti-tumor immune responses [191,192]. VEGF-mediated modulation also increases vascular permeability, creating a leaky and abnormal vasculature that limits the effective penetration of immune cells and therapeutic agents into tumors [193]. Current therapies targeting the VEGF/VEGFR pathway, including VEGF or VEGFR neutralizing antibodies and TKIs, have demonstrated substantial efficacy in patients with NSCLC [194].

#### 4.2.4. PD-L1 Expression

PD-1 is a key immune checkpoint that facilitates cancer cell immune evasion by interacting with PD-L1 on tumor cells, ultimately suppressing the tumor-killing effects of CD8+ T-cells [108]. High PD-L1 expression is a well-known predictor for excellent clinical outcomes with anti-PD-(L)1 treatment, although it is not an optimal predictive biomarker. Notably, EGFR-mutant lung cancer is characterized by an uninflamed phenotype with a high frequency of inactive tumor-infiltrating lymphocytes (TIL)s, low PD-L1 expression, and low tumor mutation burden [195,196,197,198]. In contrast to immunotherapy, studies have consistently revealed that high PD-L1 expression in the tumor tissues is associated with TKI resistance and poor clinical outcomes [198,199,200]. Interestingly, TKI treatment may affect PD-L1 expression, and the upregulation of PD-L1 has been consistently reported during the emergence of TKI resistance.

In a study on 138 patients with EGFR mutation who underwent re-biopsy after progression following EGFR-TKI failure, the proportion of patients with ≥50% PD-L1 expression increased significantly from baseline (14%) to 28% [98]. Similar results have been reported in other studies [111,112]. Various mechanisms have been suggested for PD-L1 expression elevation in EGFR TKI resistance. A study using *TCGA* database and paired NSCLC samples suggested that HGF, c-MET amplification, and the T790M mutation are involved in the upregulation of PD-L1 in NSCLC [109]. An in vitro study reported that continuous TKI treatment resulted in PD-L1 overexpression, which correlated with T-cell proliferation suppression and STAT3 and ERK1/2 pathway activation [115]. Another study using next-generation sequencing showed that mutations in the PI3K signaling pathway may lead to initial resistance to EGFR-TKIs with elevated PD-L1 levels [116]. Additionally, AKT–mammalian targeting of rapamycin pathway activation and increased BIM expression have been suggested as possible mechanisms for increased PD-L1 expression in EGFR-mutant NSCLC [117,118].

#### 4.2.5. Extracellular Matrix (ECM)

Studies emphasize the significant influence of the ECM in driving resistance to EGFR TKIs in EGFR-mutant NSCLC [201]. Integrins, crucial transmembrane receptors interacting with ECM components, play an essential role in activating signaling pathways linked to therapeutic resistance [202,203]. Integrin β1, for instance, engages with collagen-rich ECM to promote cell adhesion and trigger alternative signaling cascades such as FAK/Src and PI3K/AKT, allowing tumor cells to proliferate [204]. Similarly, integrin β3 has been associated with acquired resistance to EGFR TKIs, with its upregulation observed in resistant tumors and its inhibition being explored as a strategy to restore TKI sensitivity [205]. Collagen, a fundamental ECM component, further contributes to resistance by modifying tumor stiffness and interstitial pressure, ultimately hindering drug penetration [201]. Moreover, collagen type I has been shown to induce resistance by activating mTOR signaling through an Akt-independent mechanism, fostering the survival and proliferation of EGFR-mutant cancer cells [206].

#### 4.2.6. Adenosine Pathway

Recent studies have highlighted the critical role of the adenosine pathway in immune suppression, particularly in regulating lymphocyte activity. Within the TME, adenosine is a key modulator of immune responses [207]. Its accumulation is facilitated by ectonucleotidase CD73, which promotes adenosine production and subsequently activates immunosuppressive pathways in tumor-infiltrating immune cells [208]. Signaling through the adenosine A2A receptor has been shown to impair the cytotoxic function of CD8+ T cells and natural killer cells while promoting the differentiation of CD4+ T cells into regulatory T cells [209]. Additionally, innate immune cells within the TME express adenosine receptors, and activation of these receptors enhances the immunosuppressive functions of M2 macrophages and amplifies the effects of myeloid-derived suppressor cells [210].

Le et al. report that EGFR-mutated NSCLC, unlike EGFR wild-type, exhibits upregulation of the CD73/adenosine pathway. In the study, coculture systems with EGFR-mutant NSCLC cells demonstrated that CD73 knockdown reduced the proportion of regulatory T cells. Furthermore, treatment with an anti-CD73 antibody resulted in tumor reduction in an EGFR-mutant mouse model [211]. A study by Tu et al. reported that CD73 expression was elevated in EGFR-mutant NSCLC compared to EGFR wild-type tumors and was regulated by EGFR signaling. In a xenograft model, combination treatment with anti-PD-L1 and anti-CD73 antibodies significantly suppressed tumor growth, increased the number of tumor-infiltrating CD8+ T cells, and enhanced IFN-γ and TNF-α production by these T cells [212].

## 5. Potential Strategies for Overcoming TKI Resistance Through TME Modulation

While the previous sections discussed various TME components in relation to EGFR-TKI resistance, translating this knowledge into clinical practice remains the ultimate objective. In the current treatment setting, the goal of addressing EGFR-TKI-resistant tumors is to immediately identify actionable co-mutations. However, ongoing research on the potentially treatable components of the TME should expand future treatment options. Current research focuses on identifying biomarker candidates, elucidating the roles of targetable co-mutations in TME components, and identifying potential therapeutic targets that can be used to overcome EGFR-TKI resistance.

### 5.1. Targeting CAFs

CAFs are considered potential targets for treatment because of their role in the pathogenic crosstalk between lung fibrosis and cancer, which contributes to the fibrotic stromal components of the TME [161]. Blocking CAF-mediated signals, depleting CAF populations, and extracellular matrix remodeling may be potential therapeutic strategies for targeting the tumor-promoting functions of CAFs [153].

Preclinical models have explored the ability of antifibrotic agents pirfenidone and nintedanib to inhibit CAFs, which have shown potential as components of combination therapies aimed at slowing cancer progression [162,163]. Interestingly, pirfenidone has immunomodulatory effects as well as an anti-fibrotic effect. Studies have demonstrated that pirfenidone monotherapy can attenuate tumor growth and enhance T-cell-mediated inflammation in tumors. When combined with PD-L1 blockade, it significantly delayed tumor growth and increased survival than either treatment alone. This combination therapy promotes the expression of genes associated with innate and adaptive immune responses, resulting in increased immune cell infiltration and optimal T-cell positioning [213].

Nintedanib is a multi-target TKI that inhibits angiogenesis-related receptors, including VEGF, fibroblast growth factor (FGF), and platelet-derived growth (PDGF) [214,215]. A study demonstrated that nintedanib effectively blocked the ability of both normal and tumor-derived supernatants to facilitate aggressive cancer cell characteristics in lung cancer cell lines [216]. Although the study did not find significant changes in FGF, PDGF, or VEGF levels, it did reveal high levels of HGF, suggesting that HGF potentially facilitates cell migration and proliferation.

However, further studies should determine the potential value of CAF inhibition. One challenge in targeting CAFs is prioritizing specific subtypes within the heterogeneous CAF population and disrupting critical interactions between CAFs and other cells, which, when inhibited, can lead to the regression of TKI-resistant tumor cells.

### 5.2. Targeting TAMs

The TAM-mediated T-cell suppression is involved in EGFR-TKI resistance [121]. An in vivo study demonstrated that combining a stimulator of interferon genes (STING) agonist with osimertinib to reprogram immunosuppressive TAMs effectively induced regression of advanced tumors [122]. TAM reprogramming has emerged as a promising strategy for cancer treatment, and numerous clinical trials have explored this approach. According to a recent review, approximately 200 TAM-reprogramming agents have been investigated in more than 700 clinical trials [217]. However, strong anti-tumor efficacy is uncommon, suggesting the need to identify biomarkers for eligible patient populations and compare similar treatments earlier in the development process [217].

Reversal of the M2 phenotype to M1 is effective at overcoming gefitinib resistance. Methionine sulfoxide reductase A (MsrA) helps protect the T790M-mutant EGFR protein. When macrophages switch from the M2 to M1 phenotype, the production of reactive oxygen species (ROS) increases, lowering MsrA levels and accelerating EGFR breakdown [123]. Peng et al. developed a trastuzumab-modified liposome carrying gefitinib and vorinostat. Trastuzumab targets HER2-positive NSCLC cells, whereas vorinostat reverses the polarization of tumor-promoting M2 macrophages [123]. Similarly, Yin et al. created PD-L1-modified liposomes that delivered gefitinib and simvastatin, specifically targeting TAMs [218]. These two studies utilizing liposomes to target TAMs showed that converting macrophages from M2 to M1 increased ROS levels, inhibited angiogenesis, and boosted the release of immune-stimulating factors such as TGF-β. Trastuzumab-modified mannosylated liposomes and PD-L1-modified liposomes, both designed to deliver gefitinib, demonstrated strong anti-tumor effects and good safety profiles in NSCLC mouse models with T790M mutation. The results of these two studies indicated that targeting TAMs and modifying the TME could represent potential therapeutic strategies [219].

### 5.3. Targeting Cytokines

A study on EGFR-mutant cells suggested that combinatorial treatment with cytokines and osimertinib may be a potential treatment strategy. Ding et al. demonstrated that the combination of IL-12 with osimertinib synergistically affected tumor suppression. The authors suggested that this synergistic effect was due to increased immune cell infiltration, elevated secretion of immune-related factors, and reduced levels of immunosuppressive myeloid-derived suppressor cells [93]. However, few studies have provided preclinical or clinical evidence to support the intervention of other cytokines such as IL-6 and IL-8. To date, only a few effective combinatorial treatment strategies that directly target cytokines have been reported. A possible explanation is that compared to wild-type EGFR NSCLC, modulation of the immune microenvironment may not yield favorable outcomes. This is supported by recent studies showing that immunotherapy demonstrated limited efficacy in patients with EGFR-mutant NSCLC who progressed on EGFR-TKI therapy [220].

Although not specifically focused on patients with EGFR mutations, the NCT04691817 trial is currently enrolling participants to evaluate the safety and efficacy of tocilizumab, a monoclonal antibody targeting the IL-6 receptor, in combination with atezolizumab, for the treatment of NSCLC. This phase Ib–II trial is investigating the potential of this combination in patients with locally advanced or metastatic NSCLC who were refractory to first-line immune checkpoint inhibitor (ICI)-based therapy [221].

### 5.4. Immunotherapy

Increased activity of immune cells showing anti-tumor effects is a potential option to overcome EGFR-TKI resistance. Several key trials have been performed to identify effective immunotherapeutic options for patients with EGFR-mutant NSCLC who experience disease progression after TKI treatment.

The KEYNOTE-789 trial is a randomized, double-blind, phase III study evaluating the efficacy of pembrolizumab in combination with pemetrexed and platinum-based chemotherapy in patients with TKI-resistant, EGFR-mutant, and metastatic non-squamous NSCLC (NCT03515837). Patients with stage IV non-squamous NSCLC, confirmed DEL19 or L858R EGFR mutations, and progression after EGFR-TKI therapy were randomized 1:1 to receive pembrolizumab (200 mg every three weeks for up to 35 cycles) or placebo along with four cycles of pemetrexed and carboplatin or cisplatin, followed by pemetrexed maintenance therapy. In total, 492 patients were enrolled, and at the second interim analysis, the median PFS for pembrolizumab and chemotherapy and placebo and chemotherapy arms were 5.6 and 5.5 months, respectively (hazard ratio [HR] 0.80; 95% confidence interval [CI], 0.65–0.97; *p* = 0.0122). At the final analysis, the median overall survival (OS) in the pembrolizumab and placebo arms was 15.9 and 14.7 months, respectively (HR 0.84; 95% CI, 0.69–1.02; *p* = 0.0362). Treatment-related adverse events (TRAEs) of grade ≥ 3 occurred in 43.7% of patients receiving pembrolizumab plus chemotherapy, compared to 38.6% in patients receiving placebo plus chemotherapy. However, the addition of pembrolizumab to chemotherapy did not significantly improve PFS or OS [220]. The ongoing benefits of immunotherapy-based treatments for EGFR-mutant NSCLC following failure of EGFR-TKIs remain uncertain.

However, some studies have indicated potential benefits of immunotherapy-based treatments in patients with pretreated EGFR-mutant NSCLC. Beyond the commonly used immunotherapy options, toripalimab, a humanized IgG4κ monoclonal antibody targeting the PD-1 receptor, has demonstrated efficacy as a second-line treatment for patients with EGFR-mutant advanced NSCLC in a phase II study. The study reported an overall response rate (ORR) of 50.0%, disease control rate of 87.5%, median PFS of 7.0 months, and OS of 23.5 months [222]. In the phase II ATLANTIC trial of durvalumab in heavily pretreated patients with NSCLC, a subgroup analysis was dedicated to those with EGFR- and anaplastic lymphoma kinase (ALK)-positive disease (n = 111). For patients with PD-L1 expression ≥ 25%, the ORR was 12%, with median PFS and OS of 1.9 (95% CI, 1.8–3.6) and 13.3 (95% CI, 6.3–24.5) months, respectively. Among those with PD-L1 expression < 25%, the median PFS and OS were 1.9 (95% CI, 1.8–1.9) and 9.9 (95% CI, 4.2–13.3) months, respectively. Specifically, EGFR-positive patients (n = 97) had a median OS of 16.1 months (95% CI, 6.2–33.2) [223,224]. Additionally, the ORIENT-31 and ATTLAS trials demonstrated that adding a VEGF inhibitor to immunotherapy and chemotherapy regimens significantly improved survival outcomes [225,226].

ICI-based immunotherapy is not considered the standard treatment for treatment-naïve patients with EGFR-mutant NSCLC. The potential benefit of ICI-based therapies in patients with advanced EGFR-mutant NSCLC who have progressed on EGFR-TKIs remains uncertain. This requires further investigation, given the mixed outcomes of studies such as KEYNOTE-789 and phase II ATLANTIC [223,227]. Additionally, owing to the risk of immune-related adverse events (IRAEs) associated with immunotherapy [228,229], a cautious approach is required to minimize unnecessary risks to patients [230]. In this context, identifying the subtypes of EGFR-mutant NSCLC that are likely to benefit from immunotherapy-based treatments after progression to prior TKI therapy is crucial.

Owing to the relatively small number of patients enrolled in studies assessing the efficacy of immunotherapy in patients with EGFR mutations, no definite subgroups that would benefit from immunotherapy have been established. A retrospective study indicated that patients with prior TKI treatment and PFS < 10 months exhibited excellent response to subsequent anti-PD-1/PD-L1 immunotherapy [231]. The type of EGFR mutation and the presence of the T790M mutation were significantly associated with PFS in immunotherapy-based treatments. An indirect comparative meta-analysis showed that patients with the L858R mutation (HR 0.52; 95% CI, 0.37–0.72) and those lacking the T790M mutation (HR 0.50; 95% CI, 0.35–0.71) demonstrated significantly greater benefits from immunotherapy regimens than their counterparts [225]. High PD-L1 expression may not predict a strong response to immunotherapy, as ≥50% patients with PD-L1 expression also showed limited efficacy in a phase II trial (NCT02879994) of pembrolizumab in TKI-naïve patients [232].

### 5.5. Combination of Anti-Angiogenic and Immunotherapy

As vasculature is a key component of the TME, VEGF, a key mediator, has been targeted for intervention in various solid tumors. Multiple studies have shown that combining a VEGF inhibitor with an EGFR-TKI is significantly more effective than EGFR-TKI monotherapy in patients with advanced EGFR-mutant NSCLC.

A phase III, randomized open-label trial (NEJ026) compared erlotinib with bevacizumab to erlotinib alone in patients with stage IIIB–IV or recurrent non-squamous NSCLC with EGFR mutations. A total of 228 patients were randomly assigned in a 1:1 ratio to receive erlotinib (150 mg daily) plus bevacizumab (15 mg/kg every 21 days) or erlotinib monotherapy. The median follow-up duration was 12.4 months. At the interim analysis, the median PFS in the combination and monotherapy groups were 16.9 and 13.3 months, respectively (HR 0.605; *p* = 0.016). Serious adverse events occurred in 9% and 5% of patients in the combination and monotherapy groups, respectively [166].

ORIENT-31 is a randomized, double-blind, multicenter, phase III trial evaluating the combination of sintilimab, a PD-1 inhibitor, with the bevacizumab biosimilar IBI305, and chemotherapy in patients with EGFR-mutant non-squamous NSCLC who have progressed after EGFR-TKI therapy. A total of 444 patients who experienced disease progression after TKI were assigned to either the sintilimab plus IBI305 group or the other arms, including chemotherapy. With a median follow-up of 9.8 months (IQR 4.4–13.3), the group receiving sintilimab, IBI305, and chemotherapy showed significantly improved PFS than the chemotherapy-alone group (median PFS, 6.9 months [95% CI, 6.0–9.3] vs. 4.3 months [4.1–5.4]; HR, 0.46 [0.34–0.64]; (*p* < 0.0001). This combination demonstrated efficacy with a tolerable safety profile [226].

Similar clinical superiority has also been demonstrated with ramucirumab. A double-blind, phase III trial was conducted in patients with EGFR-mutant stage IV NSCLC without central nervous system metastasis (NCT02411448). A total of 449 patients were randomized 1:1 to receive erlotinib (150 mg/day) with intravenous ramucirumab (10 mg/kg) or a placebo every 2 weeks. The PFS was significantly longer in the ramucirumab plus erlotinib group (19.4 months) than in the placebo group (12.4 months). Serious treatment-emergent adverse events occurred in 29% of participants in the ramucirumab group and 21% in the placebo group. The most common serious adverse events were pneumonia (3%), cellulitis, and pneumothorax (2%). These results show a notable improvement in PFS with combination therapy, despite a relatively higher incidence of adverse events [233].

These clinical benefits may be associated with the role of VEGFR-2 in TKI resistance. A study by Osude et al. demonstrated that in EGFR-TKI-resistant NSCLC cell lines—specifically those harboring EGFR double mutations (L858R and T790M)—the expression of VEGF, VEGFR-2, and its co-receptor neuropilin-1 (NP-1) was significantly more elevated than that in parental or single-mutant cells. Immunofluorescence and flow cytometry confirmed increased expression of VEGFR-2 and NP-1 in TKI-resistant cells. Functional assays further revealed that combining a VEGFR-2 inhibitor with erlotinib reduced the viability of EGFR double-mutant NSCLC cells (9%) significantly more than erlotinib alone (72%) [167].

Unlike ramucirumab and bevacizumab, anlotinib inhibits multiple pathways, including VEGFR, FGFR, and PDGFR [168]. A retrospective analysis evaluated the effectiveness of combining anlotinib and ICIs in patients with NSCLC resistant to EGFR-TKI therapy. Eighty patients with lung adenocarcinoma were reviewed, with 38 receiving anlotinib plus ICIs and 42 receiving platinum chemotherapy. After a median follow-up of 15.7 months, the combination group showed significantly better PFS (4.3 vs. 3.6 months; *p* = 0.005) and OS (14.2 vs. 9.0 months; *p* = 0.029) than the chemotherapy group. Among those receiving combination therapy, 73.7% received prior treatment, with a disease control rate of 92.1%. Adverse events led to discontinuation in four patients, but most reactions were manageable and reversible [234].

The IMpower150 trial showed significant improvements in PFS and OS with atezolizumab plus bevacizumab plus carboplatin plus paclitaxel (ABCP) versus standard-of-care bevacizumab plus carboplatin plus paclitaxel in chemotherapy-naïve patients with non-squamous NSCLC [235]. In the final analysis for the subgroup with EGFR mutations, the ABCP regimen improved OS in both the overall (HR 0.60; 95% CI, 0.31–1.14) and TKI-treated (HR 0.74; 95% CI: 0.38–1.46) populations [236]. Building on the dual-targeting approach of the PD-L1 and VEGF pathways, the HARMONi-A trial evaluated ivonescimab, a bispecific antibody against PD-1 and VEGF, combined with chemotherapy, and demonstrated that dual inhibition can significantly improve PFS in EGFR-mutant NSCLC after prior TKI therapy. The trial found that the ivonescimab combination notably increased median PFS (7.1 months; 95% CI, 5.9–8.7) more than the placebo (4.8 months; 95% CI, 4.2–5.6; difference of 2.3 months; HR, 0.46 [95% CI, 0.34–0.62]; *p* < 0.001). The ORR was 50.6% (95% CI, 42.6–58.6%) for ivonescimab versus 35.4% (95% CI, 28.0–43.3%) for the placebo. Another notable finding was the pronounced PFS benefit in the T790M-positive group (HR 0.22; 95% CI, 0.09–0.54). Regarding the safety profile, ≥grade 3 adverse events were reported in 61.5% and 49.1% of patients treated with ivonescimab and placebo, primarily related to chemotherapy [237]. Considering the limitations of monotherapy in EGFR-mutant NSCLC, the addition of VEGF inhibition to immunotherapy may be a potential treatment option for patients who have progressed on EGFR-TKIs.

### 5.6. Bispecific Antibody Targeting EGFR and MET

Amivantamab is a bispecific antibody that targets both EGFR and MET in NSCLC treatment [238,239]. The unique ability of the drug to bind to two receptors simultaneously sets it apart from traditional antibodies that target only a single receptor [239]. Its mechanism of action involves three primary components: immune cell-mediated killing, receptor internalization and degradation, and the inhibition of ligand binding to both EGFR and MET receptors [240]. Amivantamab plays a significant role in immune modulation by enhancing Fc function. The drug binds tightly to Fcγ3R, mediating macrophage and NK-cell-mediated killing of cancer cells [241]. In vitro studies showed that adding isolated human immune cells notably enhanced amivantamab-mediated EGFR and MET downregulation, resulting in dose-dependent cancer cell death [242]. In this study, monocytes and/or macrophages were critically involved in Fc interaction-mediated EGFR/MET downregulation via trogocytosis [242]. This Fc-dependent macrophage-mediated anti-tumor mechanism represents a novel approach for antibody-based cancer therapies.

Promising results were obtained in a recent phase III MARIPOSA-2 trial that evaluated the efficacy of amivantamab combined with chemotherapy, with or without lazertinib, versus chemotherapy alone in patients with advanced EGFR-mutant NSCLC who experienced disease progression after osimertinib treatment [243]. In this study, PFS improved in patients receiving amivantamab-based regimens significantly more than in those receiving chemotherapy alone. Response rates were also higher with combination therapies, leading to partial or complete responses in > 50% of participants without new safety signals [243]. These results suggest that the amivantamab-based combination offers a new treatment option for overcoming resistance to EGFR-TKIs, highlighting the potential therapeutic role of immune modulation even in oncogene-addicted lung cancer.

Many combinatorial treatments, including various options such as TKIs, target-specific antibodies, VEGF inhibitors, and conventional chemotherapy, should be developed to optimize strategies for overcoming resistance to TKI treatment. Consequently, understanding the TME components in EGFR-mutant NSCLC is becoming increasingly important.

### 5.7. High-Dose Administration of TKI

One treatment strategy for overcoming TKI resistance is the administration of high-dose TKIs. In EGFR-mutant NSCLC with leptomeningeal metastases, high-dose erlotinib has demonstrated efficacy with a tolerable safety profile [244]. A systematic review evaluating the impact of higher-than-approved TKI dosing on achieving increased maximum plasma concentrations, which may lead to higher intratumoral drug levels and improved efficacy, found that high-dose intermittent TKI treatment regimens can result in elevated plasma concentrations while maintaining tolerable toxicity [245].

However, given the availability of other treatment options, preemptive exposure to doses higher than the approved TKI regimen cannot be considered a priority.

## 6. Future Perspectives

Numerous studies have emphasized the significance of each TME component in managing EGFR-mutant NSCLC. Various platforms have been used to assess the levels of these components at different transcriptional stages. For clinical implementation, much of the available research is often preliminary and requires further refinement and validation. Owing to the complexity of the TME, an intuitive approach is essential for integrating diverse information. Machine learning has recently been applied to TME analysis, prognosis prediction, and biomarker identification for predicting resistance [246,247].

Employing advanced technology can effectively organize comprehensive genetic, molecular, clinical, and spatial data of the TME, targeting specific clinical objectives, such as predicting and managing EGFR-TKI resistance. Developing a scoring system that weighs multiple clinically significant parameters related to TKI resistance could further refine this approach. Clinicians and researchers should also explore how to incorporate both genetic and non-genetic information from TME components into selecting appropriate treatment strategies. A surge in new treatment regimens has recently been introduced, necessitating a tailored approach to optimal patient care.

## 7. Conclusions

Various resistance mechanisms and targetable mutations exist in patients with NSCLC who have been treated with TKIs. Understanding the frequent resistance mechanisms specific to each targeted therapy, including those influenced by specific TME components, is essential in overcoming resistance. Additionally, predicting potential resistance to TKIs before treatment can improve patient outcomes.

## Figures and Tables

**Figure 1 biomedicines-13-00470-f001:**
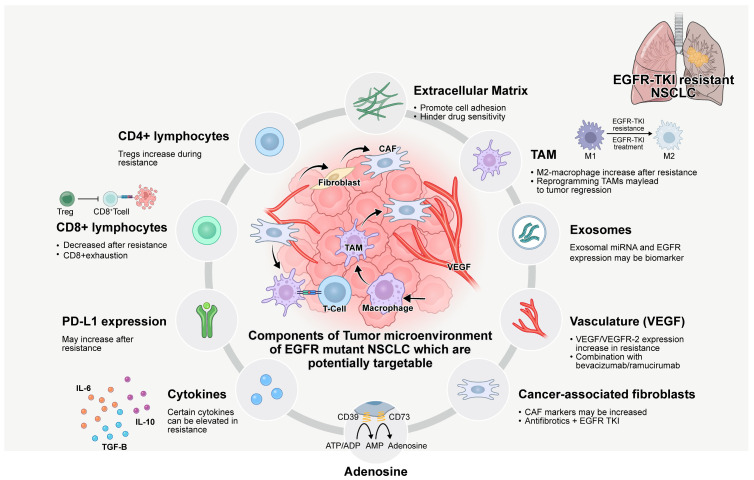
Schematic representation of key components in the tumor microenvironment (TME) of epidermal growth factor receptor (EGFR)-mutant non-small cell lung cancer (NSCLC). TME elements contributing to EGFR tyrosine kinase inhibitor (TKI) resistance. CD8+ T-cells are shown with reduced infiltration, with a specific subset (EOMES+) marked to indicate functional relevance. CD4+ T-cells (Tregs) are depicted with an increased presence near tumor cells, actively suppressing immune responses. Tumor-associated macrophages (TAMs) are illustrated with a transition from M1 (anti-tumor) to M2 (pro-tumor) phenotype, characterized by markers such as IDO1+. Cancer-associated fibroblasts (CAFs) in the stroma secrete factors such as hepatocyte growth factor (HGF) and IL-6 to promote epithelial-mesenchymal transition (EMT) and resistance in adjacent cancer cells. Cytokines, including IL-6, IL-8, and TGF-β, are shown as arrows, indicating their role in tumor growth and immune suppression. Exosomes are depicted as transferring resistance-related miRNAs between tumor and stromal cells. Abnormalities in blood vessels, i.e., leaky blood vessels, driven by vascular endothelial growth factor (VEGF) promote immunosuppressive effects. PD-L1 expression on tumor cells is elevated, inhibiting cluster of differentiation 8 (CD8+) T-cell activity and contributing to immune evasion. This schematic integrates the complex interplay of immune cells, stromal elements, and molecular signals that underpin resistance mechanisms in EGFR-mutant NSCLC. Integrins, a core component of the extracellular matrix, engage with collagen-rich ECM to promote cell adhesion and trigger alternative signaling cascades such as FAK/Src and PI3K/AKT, allowing tumor cells to proliferate. Adenosine is a key modulator of immune responses. Its accumulation is facilitated by ectonucleotidase CD73, which promotes adenosine production and subsequently activates immunosuppressive pathways in tumor-infiltrating immune cells.

## Data Availability

The datasets used or analyzed in the current study are available from the corresponding author upon reasonable request.

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
