# Peer review of "Targeting the Tumor Microenvironment in EGFR-Mutant Lung Cancer: Opportunities and Challenges"

_biomedicines, 2025, doi:10.3390/biomedicines13020470_

Round 1
Reviewer 1 Report
Comments and Suggestions for Authors
1. Targeted therapy can be discussed at the very beginning of the introduction.
2. Introductory lines on lung cancer and NSCLC are not sufficient.
3. The resistance to targeted therapy and challenges for NSCLC can be included in a separate section.
4. Note down the role of tyrosine kinase in the pathogenesis of NSCLC.
5. A section highlighting the potential tyrosine kinase inhibitors used currently for the inhibition of lung cancer and a table summarizing tyrosine kinase inhibitors used for the inhibition of angiogenesis and lung cancer can be provided
6. Write on the EFGR positive lung cancer and how does EFGR mutation induce lung cancer?
7. Significance of resistance to EGFR-TKIs should be discussed at the beginning of mechanism of resistance to EGFR-TKIs: an overview.
8. I suggest authors add the description of Receptor Tyrosine Kinases and Downstream Signaling Pathways for better understanding of the topic.
9. Why can tumor modulation be important for targeted therapy of lung cancer?
10. The authors are advised to add a list of tyrosine kinase inhibitors and their therapeutic targets used for lung cancer treatment.
11. I think that this topic is very important and therefore a separate section on the comparison of limitations and advances for the current therapeutic mode and tyrosine kinase inhibitors of NSCLC treatment.
12. The authors can discuss the dose and dose adjustments of some tyrosine kinase inhibitors used for the effective NSCLC treatment as reported by previous researchers.
Author Response
R: First of all, thank you very much for your time and effort in reviewing this article. We carefully considered all your comments and have revised our manuscript accordingly to meet your standards. During the revision process, we made substantial edits and incorporated nearly 90 new references, primarily from recent publications.
Below are the key changes made in the revised version:
#1. Section 2: Mechanism of Resistance to EGFR-TKIs – An Overview
- Added a new paragraph discussing the pathophysiology underlying EGFR mutations and the associated mechanisms of resistance.
#2. New Table
- In response to another reviewer's request, a table listing key TKIs used in NSCLC along with their respective targets has been added.
#3. New Section
- A completely new section, "Current Treatment Strategy for Overcoming EGFR TKI Resistance with Targetable Co-Mutations," has been introduced.
#4. Section 4
- Included a new paragraph discussing "Why TME Modulation Is Clinically Important."
#5. New Subsections
- Based on your comments, we have added Section 4.2.5: ECM and Section 4.2.6: Adenosine Pathways to further elaborate on the tumor microenvironment’s role in resistance.
#6. Figure
- Our figure has been extensively revised to incorporate additional information, including new components of the tumor microenvironment (TME) such as the extracellular matrix (ECM) and adenosine, along with more detailed depictions of lymphocytes and macrophages.
- Targeted therapy can be discussed at the very beginning of the introduction.
R: Thank you for your comment. For flow of the Introduction section, we added following sentence to the line 51
“Targeted therapies have significantly advanced the treatment of NSCLC by selectively inhibiting oncogenic driver mutations, leading to improved patient outcomes”
- Introductory lines on lung cancer and NSCLC are not sufficient.
R: We agree to your comment, and following sentence is added to line 47.
“NSCLC, comprising adenocarcinoma, squamous cell carcinoma, and large cell carcinoma, presents diverse molecular profiles and treatment responses.”
- The resistance to targeted therapy and challenges for NSCLC can be included in a separate section.
R: The initial Section 2 and the newly added Section 3, "Current Treatment Strategy for Overcoming EGFR TKI Resistance with Targetable Co-Mutations," discuss this issue in detail.
“3. Current treatment strategy for overcoming EGFR TKI resistance with targetable co-mutations
The current approach to overcoming EGFR TKI resistance emphasizes managing actionable co-mutations. Extensive clinical efforts aim to identify these alterations at the time of resistance. However, for patients without detectable genetic changes that allow for targeted intervention, treatment options remain limited. Some key genetic alterations that serve as targets for overcoming EGFR TKI resistance will be discussed here.
C797X mutation
Upon progression on osimertinib, approximately 15% of tumors develop on-target mutations [49], with EGFR C797X in exon 20 being the most prevalent. This mutation hampers the covalent binding of osimertinib to the EGFR kinase domain. Other notable acquired mutations include L718Q/V, G719A, and G724S in exon 18 [27]. Fourth-generation EGFR TKIs like BLU-94573 and BBT-17674 have been developed, showing promising initial data However, further investment in BLU-945 for EGFR-mutant NSCLC has been discontinued, and new treatments are awaited. Addi-tionally, preclinical studies suggest that cancer cells with the acquired C797S mutation after osimertinib therapy remain sensitive to 1G or 2G EGFR TKIs [50]. The multicenter, open-label, phase 1/2 trial (NCT05394831) is investigating JIN-A02, a fourth-generation EGFR-TKI. The results may provide insights into its potential as a treatment option for advanced NSCLC patients with C797S and/or T790M mutations [51]. Another fourth-generation TKI, BDTX-1535, showed promising outcomes in patients with re-fractory or relapsed EGFR-mutant NSCLC in a phase 2 trial (NCT05256290) [52].
MET alteration
In NSCLC, MET-dependent resistance emerges as a significant obstacle, often activated by the formation of homodimers or through trans-activation by other tyrosine kinase receptors. MET amplification contributes to resistance in approximately 50–60% of cases treated with first- and second-generation EGFR TKIs [53,54], and in 15–19% of cases involving third-generation EGFR TKIs [55]. Overcoming this resistance requires the concurrent targeting of both EGFR and MET receptors, highlighting the potential utility of anti-MET agents in combination with EGFR TKIs to achieve a more effective antitumor response [56].
To overcome EGFR TKI resistance combined with MET amplification, a combina-torial approach of EGFR TKI and crizotinib has been attempted [57][58]. Given the lim-ited antitumor efficacy of MET TKIs as monotherapy (ORR: 8.3%) in addressing ac-quired MET amplification, the combination of a MET TKI with an EGFR TKI has emerged as the most effective strategy to date [59]. This dual inhibition strategy, com-bined with osimertininb, is being explored in trials such as SAVANNAH (savolitinib) [60], INSIGHT2 (tepotinib) [61], and ORCHARD (savolitinib) [62], which have reported ORRs up to 50% and a median PFS of 5.0 months. Although this approach might im-prove outcomes compared to standard platinum-pemetrexed chemotherapy [63], its efficacy needs confirmation through ongoing trials like GEOMETRY-E (NCT04816214) and SAFFRON (NCT05261399). For osimertinib-relapsed, chemotherapy-naïve EGFR-mutant NSCLC patients, the combination of amivantamab and lazertinib has shown promising results, especially in tumors with MET overexpression by immuno-histochemistry [64]. A recent update from the INSIGHT-2 study (NCT03940703), an open-label, phase 2 trial, reported that the combination therapy of tepotinib 500 mg and osimertinib 80 mg daily achieved a significant ORR of 50.0% (95% CI 39.7–60.3) in patients with MET amplifications who had progressed following initial osimertinib treatment [65].
HER2 and HER3 alterations
The Human Epidermal Growth Factor Receptor 2 (HER2) is a tyrosine kinase re-ceptor encoded by the ERBB2 gene [66]. HER2 amplification is observed in 5% of pa-tients who develop resistance to second-line osimertinib and in 2% of cases using first-line osimertinib [67,68]. Distinct from HER2 amplification, HER2 mutations are considered to be more relevant to lung carcinogenesis, and is detected in approxi-mately 2% to 4% of patients with NSCLC patients [69,70]. Furthermore, both HER2 amplification and mutation are usually mutually exclusive with other targetable muta-tions [71,72]
One approach to overcoming HER2 amplification implicated in osimertinib re-sistance in EGFR-mutant NSCLC involves the combination of trastuzumab-emtansine and osimertinib. In a multicenter, single-arm, phase 1-2 study (NCT03784599), patients treated with this regimen showed a limited ORR of 4% and a median progression-free survival (PFS) of 2.8 months, indicating the need for alternative strategies [73].
HER3 is a member of the EGFR family, and heterodimerizes with other HER pro-teins. It is involved in cancer cell proliferation by downstream signaling through the PI3K/protein kinase B (AKT) pathway [74]. HER3, when combined with receptors such as MET, HER2, and EGFR, becomes a potent signaling entity, producing strong growth signals that can enhance resistance to targeted therapeutic interventions [75].
An antibody-drug conjugate (ADC) is a class of drug, which consists of a mono-clonal antibody linked to a cytotoxic payload via a stable chemical linker, enabling tar-geted delivery of the drug to cancer cells [76]. Initial studies indicate significant poten-tial for ADCs in treating osimertinib-resistant EGFR-mutant NSCLC. This includes ADCs targeting HER3, such as patritumab deruxtecan and BL-B01D1 (a bispecific ADC targeting EGFR and HER3), ADCs targeting TROP2 like, datopotamab deruxtecan, and ADCs targeting cMET such as telisotuzumab vedotin [77-81]. The HERTHENA-Lung02 study (NCT05338970), which included about 560 patients with EGFR mutations who had progressed during EGFR TKI therapy, indicated that patritumab deruxtecan might be effective in treating the EGFR TKI-resistant subgroup [82]. Different from other three agents, telisotuzumab vedotin requires high MET expression for efficacy, defined as a c-Met expression level of 3+ in at least 25% of tumor cells [80].
Other targetable mutations (RET, BRAF, PIK3CA)
RET fusions are also reported as acquired resistance mechanism after EGFR TKI treatment [83,84]. When compared to primary RET fusions, the proportion of CCDC6-RET in patients with acquired resistance to EGFR TKIs was higher. Addition-ally, RET fusions were more frequently linked to acquired resistance to third-generation EGFR-TKIs compared to earlier generations [85]. In fourteen patients who showed acquired RET fusions after osimertinib treatment underwent osimertinib and selpercatinib showing modest response rate, disease control rate, and median treatment duration were recorded at 50% (95% CI: 25%–75%, n=12), 83% (95% CI: 55%–95%), and 7.9 months, respectively [86]. For BRAF V600E-mediated osimertinib re-sistance, combinations such as dabrafenib and trametinib with osimertinib, as well as vemurafenib with osimertinib, have been reported in case studies involving patients resistant to osimertinib [87,88].
Recent studies indicate that the activation of the PI3K/AKT/mTOR signaling pathway contributes to the aggressive nature of lung cancer [89]. Currently, there are no established treatments that effectively target both the EGFR and PI3K/AKT/mTOR pathways simultaneously. Combinatorial approaches such as PI3K inhibitors with EGFR TKIs and PI3K/mTOR inhibitors with EGFR TKIs in overcoming EGFR TKI re-sistance mediated by the PI3K/AKT/mTOR pathway are to be investigated [90].”
- Note down the role of tyrosine kinase in the pathogenesis of NSCLC.
R: We appreciate your comment. Discussion of the role of tyrosine kinase is essential for the buildup for this manuscript. For the flow of the manuscript, we think that this can be more efficiently discussed with response to the comment 6, which will be shown later.
- A section highlighting the potential tyrosine kinase inhibitors used currently for the inhibition of lung cancer and a table summarizing tyrosine kinase inhibitors used for the inhibition of angiogenesis and lung cancer can be provided
-> Thank you for your comment. In the revised manuscript. “3. Current treatment strategy for overcoming EGFR TKI resistance” was added after the section 2. For this edit, the consecutive sections numbers had been changed.
->Regarding the table, regimens including anti-angiogenic is included in the table to be made in response to the comment 10.
- Write on the EFGR positive lung cancer and how does EFGR mutation induce lung cancer?
R: We fully agree that this explanation add strong buildup for further later arguments. For this we added following paragraph to Section 2. Mechanism of resistance to EGFR-TKIs: an overview, Line 83
“EGFR mutations play a critical role in the pathogenesis of NSCLC by driving aberrant activation of tyrosine kinase signaling pathways. These mutations lead to the constitu-tive activation of the receptor's intrinsic tyrosine kinase domain, independent of ligand binding [18].”
- Significance of resistance to EGFR-TKIs should be discussed at the beginning of mechanism of resistance to EGFR-TKIs: an overview.
R: We agree to your opinion and added following sentence to the beginning of the regarding section.
“The emergence of resistance to EGFR TKIs poses a significant challenge in the treatment of EGFR-mutant NSCLC.”
- I suggest authors add the description of Receptor Tyrosine Kinases and Downstream Signaling Pathways for better understanding of the topic.
R: Thank you for your comment. We added following sentence to Section 2. Mechanism of resistance to EGFR-TKIs: an overview, Line 83.
“The sustained activation leads to the activation of the MAPK, AKT, STAT3, and other downstream oncogenic signaling pathways, promoting processes such as uncontrolled cell proliferation, inhibition of apoptosis, angiogenesis, and metastasis [19].”
- Why can tumor modulation be important for targeted therapy of lung cancer?
R: We fully agree that explaining the importance of TME modulation in clinical aspects is essential for the later arguments. In response to your comment, we have added the following paragraph to Section 4: Role of the TME in EGFR-TKI Resistance to provide a foundation before delving into a detailed explanation of each TME component.
“Furthermore, modifying components of the TME can help restore drug sensitivity, delay resistance, and enhance treatment efficacy [99]. The TME, composed of immune cells, stromal cells, and extracellular matrix (ECM) components, plays a critical role in tumor progression and therapeutic response. For instance, targeting immune-suppressive factors such as tumor-associated macrophages (TAMs) or regulatory T cells can improve the effectiveness of targeted therapies and even synergize with immunotherapy [100]. Additionally, tumor modulation strategies allow for the co-targeting of alternative pathways that contribute to resistance against EGFR inhibitors.”
- The authors are advised to add a list of tyrosine kinase inhibitors and their therapeutic targets used for lung cancer treatment.
-> We fully agree that a table of current TKIs can be helpful for readers as background knowledge. Table 1 has been added to support the new sentence describing targeted therapy on line 51. Consecutively, old Table 1 is changed to Table 2.
Table 1. Current FDA approved targeted therapy in advanced NSCLC
Tyrosine Kinase Inhibitor |
Therapeutic Target |
Gefitinib |
EGFR (Exon 19 deletions, L858R mutations) |
Erlotinib |
EGFR (Exon 19 deletions, L858R mutations) |
Afatinib |
EGFR (Exon 19 deletions, L858R mutations, uncommon mutations like G719X, L861Q, S768I) |
Osimertinib |
EGFR (Exon 19 deletions, L858R mutations, T790M) |
Lazertinib |
EGFR (Exon 19 deletions, L858R mutations, T790M) |
Dacomitinib |
EGFR |
Alectinib |
ALK |
Ceritinib |
ALK, ROS1 |
Brigatinib |
ALK |
Lorlatinib |
ALK, ROS1 |
Crizotinib |
ALK, ROS1, MET |
Capmatinib |
MET exon 14 skipping mutations |
Tepotinib |
MET exon 14 skipping mutations |
Selpercatinib |
RET |
Pralsetinib |
RET |
Amivantamab |
EGFR, MET |
Mobocertinib |
EGFR exon 20 insertion mutations |
Dabrafenib + Trametinib |
BRAF V600E |
Encorafenib + Binimetinib |
BRAF V600E |
Abbreviations: EGFR, epidermal growth factor receptor; ALK, anaplastic lymphoma kinase; ROS1, c-ros oncogene 1; MET, mesenchymal-epithelial transition factor; RET, rearranged during transfection; BRAF, v-Raf murine sarcoma viral oncogene homolog B; V600E, valine-to-glutamate substitution at codon 600.
- I think that this topic is very important and therefore a separate section on the comparison of limitations and advances for the current therapeutic mode and tyrosine kinase inhibitors of NSCLC treatment.
R: In the newly added Section 3, "Current Treatment Strategy for Overcoming EGFR TKI Resistance with Targetable Co-Mutations," we have incorporated the following sentence to address the point you mentioned:
“The current approach to overcoming EGFR TKI resistance emphasizes managing actionable co-mutations. Extensive clinical efforts aim to identify these alterations at the time of resistance. However, for patients without detectable genetic changes that allow for targeted intervention, treatment options remain limited.”
- The authors can discuss the dose and dose adjustments of some tyrosine kinase inhibitors used for the effective NSCLC treatment as reported by previous researchers.
R: Following your advice, Section 5.7 has been added to the revised version.
“5.7. High-dose administration of TKI
One treatment strategy for overcoming TKI resistance is the administration of high-dose TKIs. In EGFR-mutant NSCLC with leptomeningeal metastases, high-dose erlotinib has demonstrated efficacy with a tolerable safety profile [234]. A systematic review evaluating the impact of higher-than-approved TKI dosing on achieving increased maximum plasma concentrations, which may lead to higher intratumoral drug levels and improved efficacy, found that high-dose intermittent TKI treatment regimens can result in elevated plasma concentrations while maintaining tolerable toxicity [235].
However, given the availability of other treatment options, preemptive exposure to doses higher than the approved TKI regimen cannot be considered a priority.”
Reviewer 2 Report
Comments and Suggestions for Authors
The processes of tumor development, invasion, metastasis, and angiogenesis activation are all significantly influenced by the tumor microenvironment. A diverse range of immunocompetent cells, fibroblasts, endotheliocytes, extracellular matrix constituents, cytokines and chemokines, and microvesicles are all part of the tumor microenvironment. Tumor cells are impacted by each of these aspects of the tumor microenvironment, and the tumor itself modifies the microenvironment to evade invasion, metastasis, progression, and immune surveillance.
Tyrosine kinase inhibitors have a significant impact on tumor processes, and a wealth of information is emerging about how tumor cells become resistant to these medications, both alone and in combination with other elements of the tumor microenvironment. As a result, reviews or meta-analyses that provide a systematic summary of the data are required.
Although this article is pertinent and makes sense, it is missing some information. For example, it does not discuss the role of the adinosine pathway (CD38/CD39/CD73) in immunity to targeted therapy or the role of extracellular matrix components (integrins, collagen) in targeted therapy.
Thirteen review articles have been published on this subject in the last five years, according to PubMed, some of which deal with the issue of immune checkpoint inhibitor exposure (Akao K, Oya Y, Sato T, Ikeda A, Horiguchi T, Goto Y, Hashimoto N, Kondo M, Imaizumi K. It might be a dead end: immune checkpoint inhibitor therapy in EGFR-mutated NSCLC. Explore Target Antitumor Ther. 2024;5(4):826-840. doi: 10.37349/etat.2024.00251.), some treatments that specifically target the epithelial growth factor receptor (Chen MT, Li BZ, Zhang EP, Zheng Q. Potential roles of tumor microenvironment in gefitinib-resistant non-small cell lung cancer: A narrative review. Medicine (Baltimore). 2023 Oct 6;102(40):e35086. doi: 10.1097/MD.0000000000035086.) and others.
The choices for targeted therapy using tyrosine kinase inhibitors and their combinations that affect the primary elements of the tumor microenvironment are compiled in this review, which is appropriate and might be of interest to readers.
The writers mostly (2/3) used materials that were no more than five years old from the date of print publication while producing the review paper. No indications of excessive self-citation were seen.
The paper's information on the mechanisms of resistance to targeted therapy in this kind of lung tumor, as well as the efficacy of monotherapy and other therapeutic approaches employed in tumor treatment, forms the basis of the article's conclusion.
It is appropriate to have illustrative material available, particularly the table, which provides a concise summary of the article's important ideas.
Remarks regarding the article: 1) It is prudent to talk about how extracellular matrix contributes to resistance to targeted therapy; 2) Figure 1 and Table 1 closely resemble the comparable illustrative content in the review article - Chen MT, Li BZ, Zhang EP, Zheng Q. Potential roles of tumor microenvironment in gefitinib-resistant non-small cell lung cancer: A narrative review. Medicine (Baltimore). 2023 Oct 6;102(40):e35086. doi: 10.1097/MD.0000000000035086. Editing is advised, especially for the drawing, and it may even be removed completely. It might be more effective to provide information regarding the involvement of different lymphocytes in sensitivity/resistance from a new line, or to compile information about tumor-filtering lymphocytes into a single block and divide them into subheadings inside.
3) A Materials and Methods section could be included, in which the primary sources for the job search, keywords for the job search, the time frame for including publications for the job search, and the inclusion/exclusion criteria could be outlined. This would enable us to recognize the authors' innovative contribution to the article's writing.
Author Response
The processes of tumor development, invasion, metastasis, and angiogenesis activation are all significantly influenced by the tumor microenvironment. A diverse range of immunocompetent cells, fibroblasts, endotheliocytes, extracellular matrix constituents, cytokines and chemokines, and microvesicles are all part of the tumor microenvironment. Tumor cells are impacted by each of these aspects of the tumor microenvironment, and the tumor itself modifies the microenvironment to evade invasion, metastasis, progression, and immune surveillance.
Tyrosine kinase inhibitors have a significant impact on tumor processes, and a wealth of information is emerging about how tumor cells become resistant to these medications, both alone and in combination with other elements of the tumor microenvironment. As a result, reviews or meta-analyses that provide a systematic summary of the data are required.
Although this article is pertinent and makes sense, it is missing some information. For example, it does not discuss the role of the adinosine pathway (CD38/CD39/CD73) in immunity to targeted therapy or the role of extracellular matrix components (integrins, collagen) in targeted therapy.
Thirteen review articles have been published on this subject in the last five years, according to PubMed, some of which deal with the issue of immune checkpoint inhibitor exposure (Akao K, Oya Y, Sato T, Ikeda A, Horiguchi T, Goto Y, Hashimoto N, Kondo M, Imaizumi K. It might be a dead end: immune checkpoint inhibitor therapy in EGFR-mutated NSCLC. Explore Target Antitumor Ther. 2024;5(4):826-840. doi: 10.37349/etat.2024.00251.), some treatments that specifically target the epithelial growth factor receptor (Chen MT, Li BZ, Zhang EP, Zheng Q. Potential roles of tumor microenvironment in gefitinib-resistant non-small cell lung cancer: A narrative review. Medicine (Baltimore). 2023 Oct 6;102(40):e35086. doi: 10.1097/MD.0000000000035086.) and others.
The choices for targeted therapy using tyrosine kinase inhibitors and their combinations that affect the primary elements of the tumor microenvironment are compiled in this review, which is appropriate and might be of interest to readers.
The writers mostly (2/3) used materials that were no more than five years old from the date of print publication while producing the review paper. No indications of excessive self-citation were seen.
The paper's information on the mechanisms of resistance to targeted therapy in this kind of lung tumor, as well as the efficacy of monotherapy and other therapeutic approaches employed in tumor treatment, forms the basis of the article's conclusion.
It is appropriate to have illustrative material available, particularly the table, which provides a concise summary of the article's important ideas.
Remarks regarding the article: 1) It is prudent to talk about how extracellular matrix contributes to resistance to targeted therapy; 2) Figure 1 and Table 1 closely resemble the comparable illustrative content in the review article - Chen MT, Li BZ, Zhang EP, Zheng Q. Potential roles of tumor microenvironment in gefitinib-resistant non-small cell lung cancer: A narrative review. Medicine (Baltimore). 2023 Oct 6;102(40):e35086. doi: 10.1097/MD.0000000000035086.
Editing is advised, especially for the drawing, and it may even be removed completely. It might be more effective to provide information regarding the involvement of different lymphocytes in sensitivity/resistance from a new line, or to compile information about tumor-filtering lymphocytes into a single block and divide them into subheadings inside.
R: First of all, thank you very much for your time and effort in reviewing this article. We carefully considered all your comments and have revised our manuscript accordingly to meet your standards. During the revision process, we made substantial edits and incorporated nearly 90 new references, primarily from recent publications.
Additionally, we appreciate your article recommendations. We reviewed the suggested literature and incorporated relevant articles that were not included in the previous version of our manuscript, with a particular focus on ECM and lymphocyte activity.
Below are the key changes made in the revised version:
#1. Section 2: Mechanism of Resistance to EGFR-TKIs – An Overview
- Added a new paragraph discussing the pathophysiology underlying EGFR mutations and the associated mechanisms of resistance.
#2. New Table
- In response to another reviewer's request, a table listing key TKIs used in NSCLC along with their respective targets has been added.
#3. New Section
- A completely new section, "Current Treatment Strategy for Overcoming EGFR TKI Resistance with Targetable Co-Mutations," has been introduced.
#4. Section 4
- Included a new paragraph discussing "Why TME Modulation Is Clinically Important."
#5. New Subsections
- Based on your comments, we have added Section 4.2.5: ECM and Section 4.2.6: Adenosine Pathways to further elaborate on the tumor microenvironment’s role in resistance.
#6. Figure
- Our figure has been extensively revised to incorporate additional information, including new components of the tumor microenvironment (TME) such as the extracellular matrix (ECM) and adenosine, along with more detailed depictions of lymphocytes and macrophages.
4.2.5. Extracellular matrix (ECM)
Studies emphasize the significant influence of the ECM in driving resistance to EGFR TKIs in EGFR-mutant NSCLC [202]. Integrins, crucial transmembrane receptors interacting with ECM components, play an essential role in activating signaling pathways linked to therapeutic resistance [203,204]. Integrin β1, for instance, engages with collagen-rich ECM to promote cell adhesion and trigger alternative signaling cascades such as FAK/Src and PI3K/AKT, allowing tumor cells to proliferate [205]. Similarly, integrin β3 has been associated with acquired resistance to EGFR TKIs, with its upregulation observed in resistant tumors and its inhibition being explored as a strategy to restore TKI sensitivity [206]. Collagen, a fundamental ECM component, further contributes to resistance by modifying tumor stiffness and interstitial pressure, ultimately hindering drug penetration [207]. Moreover, collagen type I has been shown to induce resistance by activating mTOR signaling through an Akt-independent mechanism, fostering the survival and proliferation of EGFR-mutant cancer cells [208].
4.2.6 Adenosine pathway
Recent studies have highlighted the critical role of the adenosine pathway in im-mune suppression, particularly in regulating lymphocyte activity. Within the TME, adenosine is a key modulator of immune responses [209]. Its accumulation is facilitat-ed by ectonucleotidase CD73, which promotes adenosine production and subsequently activates immunosuppressive pathways in tumor-infiltrating immune cells [210]. Sig-naling through the adenosine A2A receptor has been shown to impair the cytotoxic function of CD8+ T cells and natural killer cells while promoting the differentiation of CD4+ T cells into regulatory T cells [211]. Additionally, innate immune cells within the TME express adenosine receptors, and activation of these receptors enhances the im-munosuppressive functions of M2 macrophages and amplifies the effects of mye-loid-derived suppressor cells [212].
Le et al. report that EGFR-mutated NSCLC, unlike EGFR wild-type, exhibits upregu-lation of the CD73/adenosine pathway. In the study, coculture systems with EGFR-mutant NSCLC cells demonstrated that CD73 knockdown reduced the propor-tion of regulatory T cells. Furthermore, treatment with an anti-CD73 antibody resulted in tumor reduction in an EGFR-mutant mouse model [213]. A study by Tu et al. re-ported that CD73 expression was elevated in EGFR-mutant NSCLC compared to EGFR wild-type tumors and was regulated by EGFR signaling. In a xenograft model, combi-nation treatment with anti–PD-L1 and anti-CD73 antibodies significantly suppressed tumor growth, increased the number of tumor-infiltrating CD8+ T cells, and enhanced IFN-γ and TNF-α production by these T cells [214].
3) A Materials and Methods section could be included, in which the primary sources for the job search, keywords for the job search, the time frame for including publications for the job search, and the inclusion/exclusion criteria could be outlined. This would enable us to recognize the authors' innovative contribution to the article's writing.
R: We fully agree that a materials and methods section can be important for a review article, and we have included this section in many of our previous reviews. However, after reviewing recent articles published in our target journal, Biomedicine (e.g., https://www.mdpi.com/2227-9059/13/2/287, https://www.mdpi.com/2227-9059/13/2/299), we observed that such sections are not typically included. To maintain consistency with the journal's format, we have opted not to include a materials and methods section in this review. We kindly ask for your understanding regarding this decision.
Reviewer 3 Report
Comments and Suggestions for Authors
The review paper is detailed and consists of significant information on the tumor microenvironment in the treatment of the EGFR-mutant tumors. However, some minor revision should be conduct in the manuscript. I suggest add some figures or tables in part “4 Potential strategies for overcoming TKI resistance through TME modulation”, which could be easier for readers to understand.
Author Response
The review paper is detailed and consists of significant information on the tumor microenvironment in the treatment of the EGFR-mutant tumors. However, some minor revision should be conduct in the manuscript. I suggest add some figures or tables in part “4 Potential strategies for overcoming TKI resistance through TME modulation”, which could be easier for readers to understand.
R: First of all, thank you very much for your time and effort in reviewing this article. We carefully considered all your comments and have revised our manuscript accordingly to meet your standards. During the revision process, we made substantial edits and incorporated nearly 90 new references, primarily from recent publications.
Below are the key changes made in the revised version:
#1. Section 2: Mechanism of Resistance to EGFR-TKIs – An Overview
- Added a new paragraph discussing the pathophysiology underlying EGFR mutations and the associated mechanisms of resistance.
#2. New Table
- In response to another reviewer's request, a table listing key TKIs used in NSCLC along with their respective targets has been added.
#3. New Section
- A completely new section, "Current Treatment Strategy for Overcoming EGFR TKI Resistance with Targetable Co-Mutations," has been introduced.
#4. Section 4
- Included a new paragraph discussing "Why TME Modulation Is Clinically Important."
#5. New Subsections
- Based on your comments, we have added Section 4.2.5: ECM and Section 4.2.6: Adenosine Pathways to further elaborate on the tumor microenvironment’s role in resistance.
#6. Figure
- Our figure has been extensively revised to incorporate additional information, including new components of the tumor microenvironment (TME) such as the extracellular matrix (ECM) and adenosine, along with more detailed depictions of lymphocytes and macrophages.